# A Survey on Through-the-Road Hybrid Electric Vehicles

**Gianfranco Rizzo** [1], **Shayesteh Naghinajad** [2] **and Francesco Antonio Tiano** [1,*] **and Matteo Marino** [3]

1   Department of Industrial Engineering, University of Salerno, 84084 Fisciano, Italy; grizzo@unisa.it
2   Master Science, University of Tabriz, Tabriz 51368, Iran; shayesteh.naghinajad94@gmail.com
3   eProInn S.r.l., 84084 Fisciano, Italy; mmarino@eproinn.com
*   Correspondence: ftiano@unisa.it

**Abstract:** Hybrid Electric Vehicles (HEVs) can be divided into three categories according to how the two propulsion systems (the thermal and the electric ones) supply the driving torque to the vehicle. When the torque is supplied only by an electric propulsion system, while the heat engine takes care of generating the electricity needed to operate the system, it is called a hybrid-series. Conversely, when both propulsion systems provide torque, the vehicle is identified with parallel hybrid wording. Among the parallel hybrids there is a particular configuration called Through-the-Road (TTR). In this configuration, the two propulsion systems are not mechanically connected to each other, but it is precisely the road that allows hybrid propulsion. This architecture, dating back to the early twentieth century, is still used by several manufacturers and carries with it peculiar configurations and control methods. It is also a configuration that fits well with the transformation of conventional vehicles into a hybrid. The paper presents a survey of the TTR HEV solution, evidencing applications, potentialities and limits.

**Keywords:** hybrid; hybridization; HEV; Through-the-Road; TTR; automotive; state-of-art; sustainable mobility

## 1. Introduction

Currently, road vehicles are responsible for large fractions of global oil consumption and of $CO_2$ emissions. Achieving a more efficient, flexible and environmental safe transport is now a widely pursued goal. The transportation sector is facing a challenge in the form of strict emission standards which needs to be complied with by all new vehicles [1,2].

In recent years, there has been an increase in the number of electric vehicles in the world, motivated by the choice of consumers and a fully transitioned automotive industry. Nevertheless, several problems remain to be overcome before a mass transition to electric cars:

- a notable increase in electricity production would be required, and obtained from renewable sources, otherwise the $CO_2$ reduction target would be lost [3];
- the charging power of the batteries is still too low, and therefore the charging times are much higher compared to those referring to motor vehicles;
- the electricity grid could collaps by a general and uncontrolled recourse to the recharging of electric and plug-in vehicles, before major adjustments to the network structure and its "philosophy" (Smart-Grid) [4,5].

One of the most suitable short-term solution to this problem consists in the increasing use of Hybrid Electric Vehicles (HEV), in the meantime that the several issues preventing a mass diffusion of

electric or Fuel-Cell Electric Vehicles (FCEV) will be solved [6–11]. HEVs can combine the advantages of the purely electric vehicles, in particular zero local emissions, with the advantages of the ICE-based vehicles, namely high energy and power density. HEVs can profit from various possibilities for improving the fuel economy with respect to ICE-based vehicles. In principle, it is possible to downsize the engine and still fulfill the maximum power requirements of the vehicle. Additionally, recover some energy during deceleration instead of dissipating it in friction braking [7].

Among the HEV configurations, the Through-the-Road (TTR) architecture can be found. As will be seen in more detail below, a HEV in TTR configuration provides that one axle of the vehicle is driven by the internal combustion engine, while the other axle is driven by an electric propulsion system [7]. This technical solution, currently in use on various ranges of hybrid vehicles, is widely studied in the literature also for the possibility of being used for the conversion of conventional vehicles into hybrid vehicles [12].

This work proposes a survey on the State-of-the-Art of the peculiar TTR HEV architecture. Section 2 introduces the hybrid vehicles configurations, with a focus on TTR HEVs, giving in addition a wide overview on TTR HEV models produced by major car manufactures. Section 3 is the core of this study, where state-of-the-art of research and applications are widely exposed. In particular, both technological choices, such as vehicle conversion, and control strategies are presented. An analysis of patents regard the TTR HEV technology is presented as well. Lastly, Section 4 summarizes the work and gives hints on the future of this technology.

## 2. TTR HEV Architecture

As is well known in the literature and in everyday life, a HEV is a vehicle that is characterized by two propulsion systems: an internal combustion engine and an electric mover (which can work also as a generator, for recovering the braking energy) [7].

Hybrid vehicles can have three different configurations [7]:

- parallel hybrid, where both propulsion systems operate simultaneously;
- hybrid series, where the motion of the vehicle is determined solely by the electric motor and the heat engine is used to recharge the battery;
- series-parallel (or combined) hybrid, where there is a combination of the two previous configurations.

One of the possible ways for realizing a parallel hybrid electric vehicle is the so-called Through-the-Road (TTR) architecture. In a TTR HEV the two propulsion systems are mounted on separate axles. Hence, the term TTR refers to the power coupling scheme between the internal combustion engine (ICE) and electric movers, which is not through some mechanical device but through the vehicle itself, its wheels and the road on which it moves [13]. Consequently, the powertrain architecture of the TTR HEV consists of an Internal Combustion Engine (ICE) that drives the front wheels, and an electric propulsion system driving the rear wheels [14]. The electric mover can actually be part of the wheel itself, called an in-wheel motor (IWM) [15,16]. Possible architectures for a TTR HEV are shown in Figure 1.

The major difference between a TTR HEV and a conventional parallel HEV is in the nonexistence of physical link between the mechanical and electrical drive. So the complex torque coupling device becomes unessential and leads to a simpler and cheaper implementation of parallel HEVs. The 4-wheel drive capability, which makes vehicle more stable with exceptional acceleration, can be come to account as an advantage of this configuration [17].

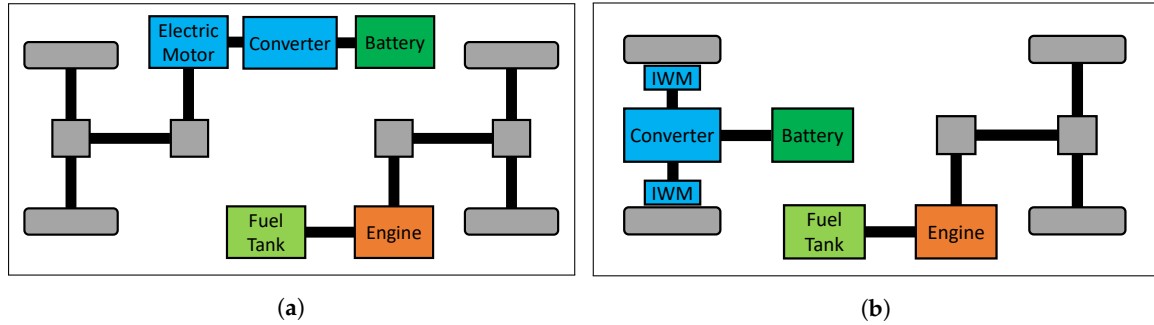

(**a**) (**b**)

**Figure 1.** Possible architectures for a TTR HEV. (**a**) One axle is driven by the internal combustion engine, while the other is driven by an electric motor; (**b**) Electric motors are mounted directly inside the wheels (In-Wheel Motor).

Far from being a naive solution, TTR architecture is also used by major car manufacturers such as Peugeot, Citroen, Volvo, BMW and Honda. The TTR HEV offered by these manufacturers are shown in Table 1.

**Table 1.** Commercial hybrid vehicles using a TTR architecture.

| OEM | Model | Years | Description |
|---|---|---|---|
| Peugeot | 3008<br>508<br>508 RXH | 2012–2016 | Equipped with *Hybrid4* powertrain which combines a 120 kW 2.0 liter diesel engine with a 28 kW electric motor on the rear axle. Another electric motor is in the engine compartment and serves as starter for the start-stop system. |
| Citroen | DS5 | 2011–2018 | |
| Volvo | XC90 Recharge<br>XC60 Recharge<br>XC40 Recharge<br>V90 Recharge<br>V60 Recharge<br>S90 Recharge<br>S60 Recharge | Since 2019 | Volvo's PHEVs range are equipped with *AWD Twin Engine* powertrain which combines a gasoline engine, which drives the front wheels, with an electric motor which acts on the rear wheels. Three powertrain configurations are available, called T5, T6 and T8, and feature an ICE of 132, 186 and 223 kW and an electric motor of 60, 65 and 65 kW, respectively. |
| BMW | Series 2 xe<br>Series 3 xe<br>Series 5 xe<br>Series 7 xe<br>X1<br>X3<br>X5 | Since 2015<br>Since 2016<br>Since 2017<br>Since 2019<br>Since 2015<br>Since 2017<br>Since 2020 | Equipped with a powertrain composed of an ICE that drives the front wheels and an electric motor acting on the rear wheels. Several configurations in terms of power of both ICE and electric motor are available. |
| | i8 | 2014–2020 | The powertrain includes an electric motor located in the front axle powering the front wheels rated 96 kW until 2018 and 105 kW from 2018 and a turbocharged 1.5-liter gasoline engine driving rear wheels rated 164 kW. |
| Honda | NSX 2nd gen | Since 2016 | Equipped with *Sport Hybrid SH-AWD* powertrain composed of a 3.5 liters VTEC 378 kW engine combined with a 35 kW electric motor placed between the engine and the gearbox which drive the rear wheels, and two 27 kW electric motors placed on the front axle. |

## 3. State of the Art of Research And Applications

### 3.1. Vehicle Conversion

Another advantage of the TTR HEV architecture is the prospect to retrofit any conventional ICE vehicles and transform them into HEVs [12,18]. Upgrading conventional vehicles to Hybrid Electric Vehicles (HEVs) can represent a viable and feasible way to reduce fuel consumption and

emissions, particularly in urban areas, avoiding a premature and massive scrapping of conventional cars. The "reuse" of the car, in a Life Cycle Assessment (LCA) perspective, allows reducing the impact on $CO_2$ and energy consumption due to a premature dismissal of the car [19–21]. Some cases reported in literature are presented in the following.

3.1.1. University of Salerno (Italy)—LIFE-SAVE (Solar Aided Vehicle Electrification) Project

A proposal of car hybridization, also including solar recharging by PV cells, has been formulated and patented at the University of Salerno (Italy) [19,22–27]. The hybridization kit consists in two in-wheel motors, an additional Lithium-ion battery, solar photovoltaic modules installed on the vehicle body and a control unit for the electric powertrain. A schematic depiction of the hybridization kit is shown in Figure 2. A FIAT Grande Punto, shown in Figure 3, has been converted in a solar assisted TTR parallel hybrid electric vehicle.

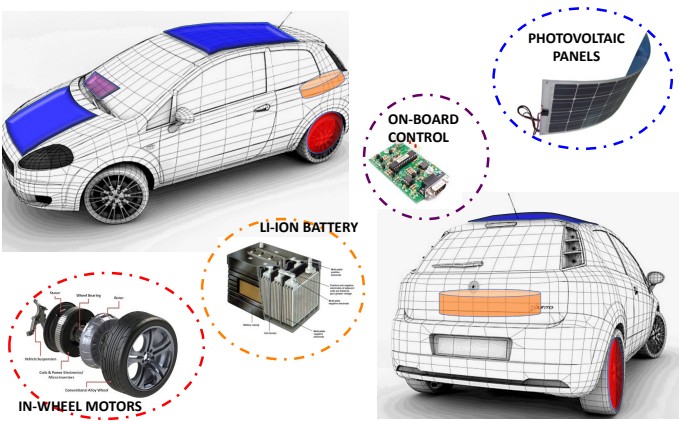

**Figure 2.** Schematic depiction of the solar hybridization kit.

The solar hybridization allows enhancing energy savings and emissions reduction, due to free solar recharge and partial recovery of braking power. Although the benefits in terms of fuel savings would be lower than on a native hybrid, due to the constraints posed by the TTR structure, and by the absence of downsizing effects, the addition of electric propulsion may offer further advantages, such as:

i　　the enhancement of vehicle power and performance (in particular acceleration),
ii　　increase in vehicle reliability, due to presence of two propulsion devices,
iii　　better driveability, due to the possibility of exploiting advanced vehicle control schemes.

Studies performed by a simulation analysis, adopting a set of models experimentally validated, have shown that the solar hybridized vehicle can achieve significant reduction of fuel consumption, up to 20% in typical urban driving [25,27]. The results of an analysis performed by Dynamic Programming have allowed to confirm that the major benefits are achieved in urban driving (FUDS cycle), and to obtain a trade-off between the level of complexity adopted for hybridization (use of Drive by Wire) and fuel savings [27]. Real-time detection of active gear and of neutral gear conditions, without using additional sensors and making use only of low frequency OBD data, is achieved by proper real-time models [26]. A study performed with a forward vehicle model including driver simulation has shown that possible delays in the control and actuation chain do not impair significantly drivability, and have evidenced the existence of a limiting value for Power Split value when Drive by Wire is not adopted [19]. A survey conducted over more than a thousand potential users has shown good attitude toward this solution, with quite high level for the "Intention to adopt" [28]. The project has been financed by the European program LIFE, with the participation of four Italian partners, to produce prototypes ready for industrialization (TRL = 9). Details at the website www.life-save.eu.

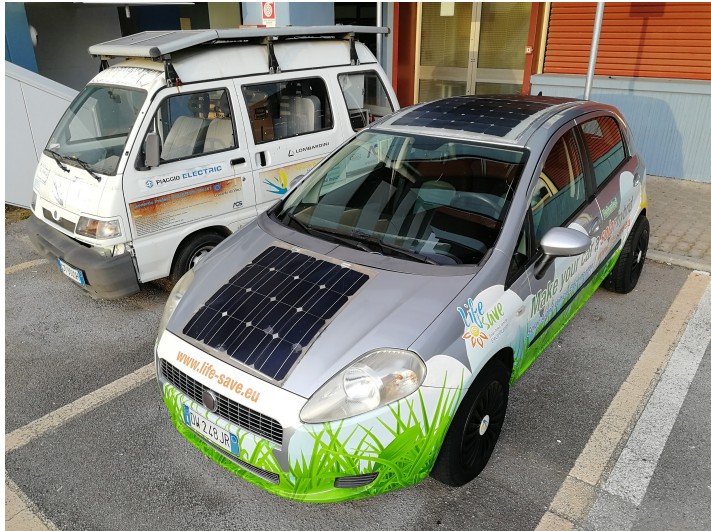

**Figure 3.** Hybrid Solar Vehicle prototype realized by University of Salerno on a FIAT Grande Punto.

### 3.1.2. Universiti Teknologi Petronas (Malaysia)

A project to retrofit a Proton WAJA into a hybrid vehicle has been proposed by a team at the University Technology Petronas (UTP) (Malaysia) in collaboration with an industrial partner, H2E Technologies Sdn. Bhd. [29]. Similarly to what has been developed by the University of Salerno, this proposal also involves the use of IWMs and the addition of a battery placed in the trunk of the car. The attach of IWMs to the rear split-axle leads to the modification of the original vehicle structure [16].

The team also proposes to replace the cable-driven engine throttle with an electronic device for two main reasons. First, this modification disengages the driver's accelerator pedal command from the engine throttle; secondly, this modification allows to process the driver's torque request and split it between the internal combustion engine and the electric motors [30].

### 3.2. Impact of ICE Sizing

The impact of the ICE sizing on fuel economy and $NO_x$ emission improvements which can be obtained with the hybrid technology are investigated in [31]. For simulation, two TTR parallel HEVs—with an architecture depicted in Figure 1a—equipped with ICEs characterized by a different maximum output power, and a conventional vehicle used as a reference, were investigated. The two TTR HEV have been modeled with a 30% (TTR30) and a 50% (TTR50) degree of hybridization, respectively. Considering the efficiency of the ICE and the Electric Motor (EM), all the data were normalized, due to the rational property reasons. In order to obtain the same vehicle performance as the one of the conventional vehicle, all factors like maximum power, maximum vehicle velocity, 0–100 km/h acceleration time and gradeability are considered. An additional constraint has been introduced with respect to the conventional vehicle, that is, the All-Electric Range (AER). This is the maximum distance that the vehicle can cover in pure electric, and it determines the minimum battery capacity. Table 2 expresses the characteristics of the conventional vehicle.

The layout of the two vehicles is optimized through an optimal-layout tool which generates a population of potential optimal design solutions specifying the size of the electric motor, of the battery pack and the gear ratio of the front and rear final drives. Considering the powertrain cost, the solution can be optimal if it minimizes the cost. The powertrain cost covers the cost of the ICE, of the EM, of the battery pack and of the consumed fuel to cover 160.000 km over a representative mission constituted by a combination of several driving cycles, including the NEDC, FTP and the three Artemis cycles [31]. In order to fulfill the maximum power constraint, the minimum power of the battery should be almost equal to the difference between the power of the downsized engine of the hybrid vehicle and maximum power of the conventional vehicle.

**Table 2.** Main specifications of the conventional vehicle [31].

| Specification | Conventional Vehicle |
|---|---|
| Vehicle Mass (kg) | 1786 |
| Chassis Mass (kg) | 1500 |
| Drag Coefficient (-) | 0.3 |
| Front Area (m$^2$) | 2.45 |
| Tyre radius (m) | 0.327 |
| Max. speed (km/h) | 224.5 |
| 0–100 Acceleration time (s) | 7.6 |

The sizes of final components are demonstrated in Table 3. It has been proved that the battery capacity depends on the energy content demanded to fulfill the AER constraint of the vehicle. In addition, the battery size relies on the maximum power of the EM, which is related to the maximum power limitation of the powertrain. Moreover, the minimum allowable size of the EM depends on the restrictions on 0–100 km/h acceleration time and maximum vehicle velocity.

The sizes of final components are demonstrated in Table 3. It has been proved that the battery capacity depends on the energy content demanded to fulfill the AER constraint of the vehicle, also size of the battery relies on the maximum power of the EM, which is related to the maximum power limitation of the powertrain. Moreover, the minimum allowable size of the EM depends on the restrictions on 0–100 km/h acceleration time and maximum vehicle velocity.

**Table 3.** Specifications of ICEs, EMs and batteries obtained from the Optminal Layout Tool [31].

| Category | Specification | Conventional Vehicle | TTR30 | TTR50 |
|---|---|---|---|---|
| Vehicle | Mass (kg) | 1786 | 1798 | 1813 |
| | Max. speed (km/h) | 224 | 229.5 | 229 |
| | 0–100 Acceleration time (s) | 7.6 | 7.6 | 7.6 |
| | Degree of hybridization (%) | - | 23.4 | 49.3 |
| ICE | Max. Power (kW) | 140 | 100 | 70 |
| | Displacement (l) | 2.9 | 1.7 | 1.3 |
| | Mass (kg) | 210 | 161 | 130 |
| | Spec. Power (W/kg) | 667 | 621 | 538 |
| Battery | Type | - | Lithium-Ion | |
| | Max. Power (kW) | - | 32.7 | 69 |
| | Max. Current (A) | - | 120 | 120 |
| | Cell Capacity (Ah) | - | 5.5 | 5.5 |
| | Cell Voltage (V) | - | 3.6 | 3.6 |
| | Spec. Power (W/kg) | - | 1400 | 1400 |
| | Mass (kg) | - | 23.3 | 37.8 |
| EM | Type | - | Permanent magnet | |
| | Max. Power (kW) | - | 32.5 | 70 |
| | Spec. Power (W/kg) | - | 1850 | 1850 |
| | Mass (kg) | - | 17.6 | 49 |
| Final Drive | Front Speed Ratio (-) | 3.1 | 3.25 | 3.25 |
| | Rear Speed Ratio (-) | - | 4 | 4 |

Figure 4 shows the weight distribution of the powertrain components for two HEVs and reference conventional one. The chassis mass and the gearbox mass has been kept stable for the three mentioned vehicles. Due to the fact that the ICE is mechanically connected to the wheels, the gearbox cannot be omitted in the hybrid vehicles. Both hybrid vehicles are heavier than the conventional one, but the mass increment is very small with 12 kg and 27 kg for TTR30 and TTR50, respectively. This is due to the power density of the battery, which is undeniably high for both vehicles. The mass of the EM was found to be approximately 75% of that of the battery. The TTR50 is slightly penalized in

terms of weight, as the specific power of the ICE installed in TTR50 (538 W/kg) is lower than one of TTR30 (621 W/kg), while the specific power of both the EM and battery installed in these layouts remains stable.

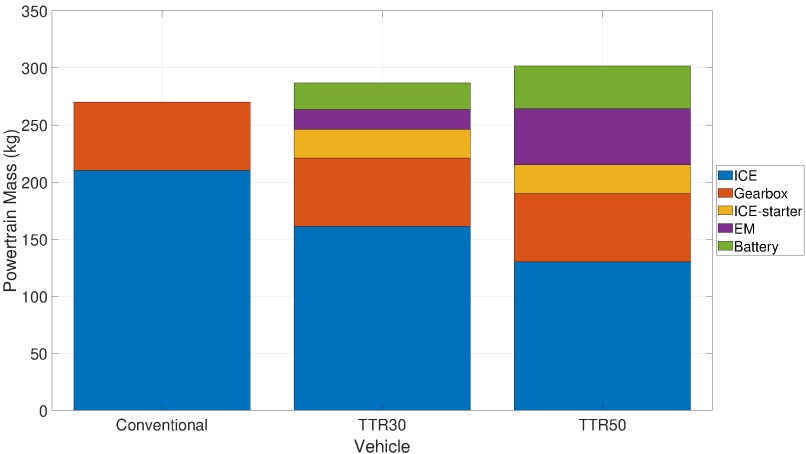

**Figure 4.** Weight distribution of the powertrain components optimized with the Optimal Layout Tool [31].

*3.3. Control of TTR HEV*

Alongside the analysis of architecture and the size of components, the critical factor in a TTR HEV is the vehicle control strategy [15]. The satisfying control strategy needs essentially to achieve the control objectives such as: satisfy the driver's power demand, reduce fuel consumption and emission, sustain a rational level of battery SOC for self-sustaining operation (no external charging), and regain the optimal amount of braking energy [32,33].

3.3.1. ON/OFF Strategy

The "ON/OFF" control strategy (also called "Thermostat" or "Bang-Bang") was firstly proposed for a series hybrid vehicle and after that is extended to the power flow control in a parallel HEV vehicle. Demanded torque of engine, quality of engine operation and launch speed (i.e., a pre-determined minimum vehicle speed) represent the main considerable factors of this strategy. In the condition in which vehicle demanded speed is greater than the launch speed, the ON/OFF strategy allows the vehicle to work in only engine, only electric motor and hybrid mode. On the contrary, whether the vehicle demanded speed is lower than launch speed only engine or only motor modes are available [34,35]. Figure 5 shows the operating modes of the strategy.

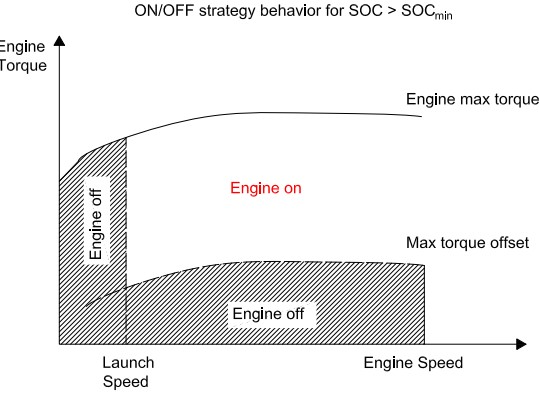

**Figure 5.** The operating modes of the ON/OFF strategy [34].

### 3.3.2. Electric Assist Control Strategy

The Electric Assist Control Strategy (EACS) represents a commonly used rule-based control strategy [36]. It uses the ICE as the main power source and the battery to assist the ICE. The EACS operates with the battery when the ICE works inefficiently or the load demand is beyond the ICE maximum power. However, since there are many parameters to tune in the EACS, its design process is time and effort consuming. On the other hand, the ICE produces additional power to charge the battery when the SOC is low. The control rules of the EACS are presented in Figure 6, where $V_{limit}$, $r_{min}$ and $r_{off}$ are three tunable parameters determining the thresholds for the ICE activation, while $P_{chrg}$ is the additional power delivered by the ICE to charge the battery in addition to the propulsion load when the SOC value is low:

$$P_{chrg} = r_{chrg} \cdot \left( \frac{SOC_L + SOC_U}{2} - SOC \right) \cdot \omega_e \tag{1}$$

where $SOC_L$ and $SOC_U$ are the lower and upper limit of battery SOC, respectively, $SOC$ is the actual battery SOC, $\omega_e$ is the ICE rotational speed and $r_{chrg}$ is a parameter to be tuned.

Operative SOC limits of HEVs battery depends on the battery typology and the vehicle control strategies. In general, in order to preserve battery life and safety, the SOC is limited between the 20% and the 90% of charge [37], similarly to what happens for batteries in micro-grid applications [38,39].

When the battery SOC is in the desirable operation region, i.e., $SOC_L \leq SOC \leq SOC_U$, the EACS works with the load-following mechanism.

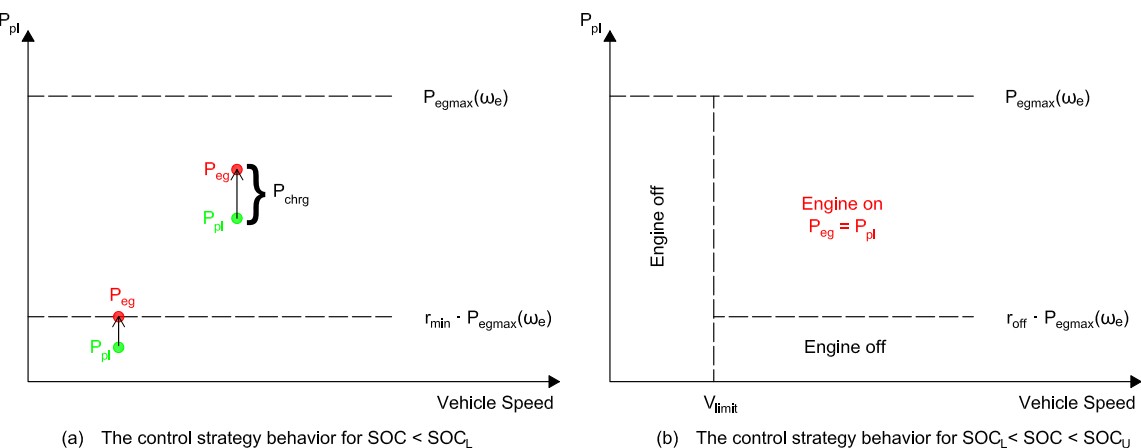

(a) The control strategy behavior for SOC < $SOC_L$　　　　　(b) The control strategy behavior for $SOC_L$< SOC < $SOC_U$

**Figure 6.** The operating modes of the EACS [36].

### 3.3.3. Fuzzy Logic

The fuzzy rule-based method, due to the fuzzification process, is relatively robust and able to deal with the nonlinearity and uncertainty [40]. In Reference [41] a fuzzy control system in terms of the engine optimal efficiency map on the output torques of the engine and motor has been applied. The NREL's ADvanced VehIcle SimulatOR (ADVISOR) is used in order to simulate vehicle considering two targets:

1.　Maximizing the efficiency of the engine;
2.　Marking torque-speed operating points to achieve minimum fuel consumption.

In order to simulate the fuzzy strategy in ADVISOR, Shaheb 2 (a parallel TTR HEV based on a KIA Pride) has been designed and manufactured at the University of Kashan [18]. A simplified schematic of the Shaheb 2 architecture is shown in Figure 7. Definition of optimum points to reduce

the fuel consumption is the main objective. The application of this fuzzy control resulted in an average fuel converter efficiency of 29.43% on the UDDS driving cycle. Table 4 indicates the vehicle and engine technical data of Shaheb 2.

In this paper, a multi-objective approach with two fuzzy controllers for torque and for state of charge (SOC) was adopted. In fuzzy torque controller, the normalized SOC is assumed as input, and engine torque as output. In SOC fuzzy controller, normalized SOC and its derivative are assumes as input, and additional torque as output: this term is then summed to the torque value provided by the first controller. The recourse to the SOC controller has also the purpose to determine a suitable battery depletion pattern. An adjustable gain allows matching the desired vehicle power. The membership functions for the two controllers make use of seven levels: positive big (PB), positive medium (PM), positive small (PS), zero (ZR), negative small (NS), negative medium (NM), and negative big (NB).

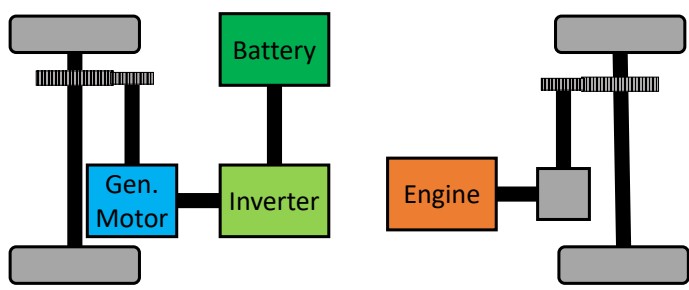

**Figure 7.** Block diagram of Shaheb 2 architecture [18].

**Table 4.** Technical data of Shaheb 2 [18].

| Vehicle and Engine | |
|---|---|
| Engine rated power (kW) | 45 |
| Maximum speed in electrical mode (km/h) | 100 |
| Range in electrical model (km) | 25 |
| Grade ability (%) | 45 |
| Acceleration (s) | 18 |
| **Electrical** | |
| Electric motor power (kW) | 22 @ 3000 rpm |
| Nominal voltage (V) | 144 |
| DC-Bus voltage (V) | 192 |
| Maximum speed (rpm) | 6000 |
| Rated torque (Nm) | 75 |
| Efficiency % | 95 |
| Inverter power (kVA) | 45 |
| Inverter hardware | DSP TMS320LF2407A |
| Control scheme | Space phasor vector controlled |
| Battery type | Lithium-ion polymer |
| Battery cell voltage (V) | 14.8 |
| Battery nominal capacity (mAh) | 10,500 |

In Reference [42] a fuzzy control system is applied as a main control block with a pair of membership functions to help the power flow controller to choose the adequate power diffusion by the hybrid drivetrain based on accessible resources in real time. In order to get the minimum fuel consumption for the desired trip, the Energy Management Strategy (EMS) operates in hybrid mode blended control strategy. The TTR equipped with in-wheel motors (IWM) in the rear wheels, with the same architecture depicted in Figure 1b, was simulated in MATLAB as a Simulink model. The reference model which is used for comparison is a series–parallel HEV with rule-based EMS. The vehicle switches into hybrid mode when the needed power to the wheel oversteps the Power Threshold, $P_{th}$, line, it means that the level of $P_{th}$ will be the determining multiplier for the mode selection of the EMS

controller [43]. The power flow in the TTR HEV is decided based on current vehicle speed and the Global Discharge Rate (GDR) value derived from the current State-Of-Charge (SOC) of the battery and remaining trip distance [42]. Four different driving cycles are investigated and comparisons against other published models are equally encouraging, especially on high-average-speed drive cycles with up to 19.8% improvements in fuel consumption. Related results have been indicated in Table 5 for different values of the GDR:

- Very Very Low (VVL), 0%/km;
- Very Low (VL), 1%/km;
- Low (L), 2%/km;
- Medium Low (ML), 3%/km;
- Medium (M), 4%/km;
- Medium High (MH), 5%/km;
- High (H), 6%/km;
- Very High (VH), 7%/km;
- Very Very High (VVH), 8%/km.

**Table 5.** Fuzzy rules for Power Threshold ($P_{th}$) mapping [42].

| $P_{th}$ for Vehicle Speed | GDR (%/km) | | | | | | | | |
|---|---|---|---|---|---|---|---|---|---|
| | VVL (0) | VL (1) | L (2) | ML (3) | M (4) | MH (5) | H (6) | VH (7) | VVH (8) |
| Slow | 3937 | 5250 | 6562 | 7957 | 9680 | 10,459 | 10,459 | 10,459 | 10,459 |
| Med | 7684 | 8655 | 10,455 | 11,341 | 13,728 | 16,740 | 19,095 | 19,095 | 19,095 |
| Fast | 7523 | 10,457 | 12,908 | 15,094 | 17,106 | 20,125 | 21,171 | 23,556 | 32,157 |

Unit: Watts.

Improvement thanks the proposed strategy are summarized in Table 6. From the analysis of the text it is not clear or evident whether the same SOC variation in battery was recorded in the two cases.

**Table 6.** Fuel consumed comparison between reference and proposed model in [42].

| Driving Cycle | Reference Model | Proposed Model | Improvement |
|---|---|---|---|
| ECE R15 | 0.1652 L | 0.1300 L | 21.0% |
| EUDC | 0.6581 L | 0.2974 L | 54.8% |
| NEDC | 1.3310 L | 0.8242 L | 38.1% |
| HWFET | 1.4220 L | 0.5410 L | 62.0% |

The applied control strategy makes the proposed TTR HEV indicates undesirable performance on lower average speed cycle like the NEDC drive cycle despite higher average speed drive cycle such as the HWFET. Considering ICE size which is relatively bigger than conventional ones, the shortage in fuel consumption on NEDC driving cycle is undeniable since the bigger ICE leads to higher fuel consumption even though if it is idle.

The team that design the Shaheb 2 vehicle [18] made a comparison between the fuzzy logic and the ON/OFF strategy for the control of the vehicle. Fuzzy logic approach resulted in a fuel economy 27% better than the one obtain with the use of the ON/OFF strategy. Tables 7 and 8 report a summary of the comparison between ON/OFF and Fuzzy strategies on the UDDS cycle.

**Table 7.** Emissions and fuel consumption results [18].

| Control Strategy | Fuel Economy (L/100km) | CO Emission (g/km) | HC Emission (g/km) | NOx Emission (g/km) |
|---|---|---|---|---|
| **ON/OFF** | 4.1 | 3.954 | 0.426 | 0.372 |
| **Fuzzy** | 3.7 | 1.582 | 0.418 | 0.350 |

<p align="center">**Table 8.** Performance characteristics results [18].</p>

| Control Strategy | Time 0–100 km | Distance after 5 s | Time at 400 m | Max Speed (km/h) |
|:---:|:---:|:---:|:---:|:---:|
| **ON/OFF** | 10 | 50.6 | 17.2 | 179.7 |
| **Fuzzy** | 9.6 | 52.8 | 17.0 | 180.8 |

### 3.3.4. Torque Leveling Threshold Changing Strategy (TTS)

In Reference [44,45] the Torque Leveling Threshold Changing Strategy (TTS) is proposesed for a TTR HEV with IWMs architecture as depicted in Figure 1b. This new fundamental concept of torque leveling is applied and compared with the Electric Assist Control Strategy (EACS) which one of the prevalent strategies for HEVs control and is based on the load following approach. The idea behind this mechanism is to run the ICE at a constant torque during the time the engine is running. Consequently, the efficiency of the engine can be guaranteed. The other advantage of the TTS is that it can adopt the threshold changing mechanism to operate the HEV in a charge-sustaining method.

In the TTS strategy, the powertrain works in net electric mode at low power demand, considering the fact that the low and medium power demands are identified by a battery charge and engine speed dependent threshold, the powertrain enters hybrid action with the ICE operating with a constant torque at medium power demands. So the ICE operates at its maximum power at higher loads or when the battery charge falls below a lower threshold. In other words, the vehicle is working in pure electric mode at low power demand, while for medium and high power demand the internal combustion engine is turned on and gives the required extra power.

The Ricardo Wave CFD engine simulator [46] was used to obtain the engine map from experimentally validated simulations in order to investigate the fuel consumption. The one used in simulation is a 2 liters petrol engine with a peak power $P_{max}$ of 120 kW and a peak torque $T_{max}$ of 300 Nm.

Fuel consumption $m_f$ and equivalent fuel consumption $m_{efc}$ achieved with the TTS application on four driving cycles WL-L (low speed), WL-M (medium speed), WL-H (high speed) and WL-E (extra-high speed) from the Worldwide harmonized Light vehicles Test Procedures (WLTP) and their deviation from results obtained with an optimal control strategy found via Dynamic Programming (DP) are shown Table 9 [45].

<p align="center">**Table 9.** Fuel Economy results for TTS [45].</p>

| Drive Cycle | SOC$_{final}$ | $m_f$ (kg) | $m_{efc}$ (kg) | $\Delta_{DP}$ |
|:---|:---:|:---:|:---:|:---:|
| WL-L | 0.6512 | 0.0949 | 0.0944 | +2.28% |
| WL-M | 0.6493 | 0.1596 | 0.1599 | +1.14% |
| WL-H | 0.6494 | 0.2586 | 0.2588 | +2.01% |
| WL-E | 0.6487 | 0.4122 | 0.4127 | +1.33% |

Analogously, Figure 8 shows a comparison of EACS and TTS expressed as percentage difference compared to the optimal DP control. It can be noted that the EACS strategy does not show a satisfying result. On the same four driving cycles, the fuel economy obtained with the application of EACS is higher than the DP results by 6.02% on average (range 3.19%–10.08%). On the contrary, the TTS is able to achieve an impressive fuel economy, which is higher than the DP of only 1.14%–2.28% [45].

Additionally TTS can be easily implemented for any kind of vehicle for which other control systems can not been easily implemented [44].

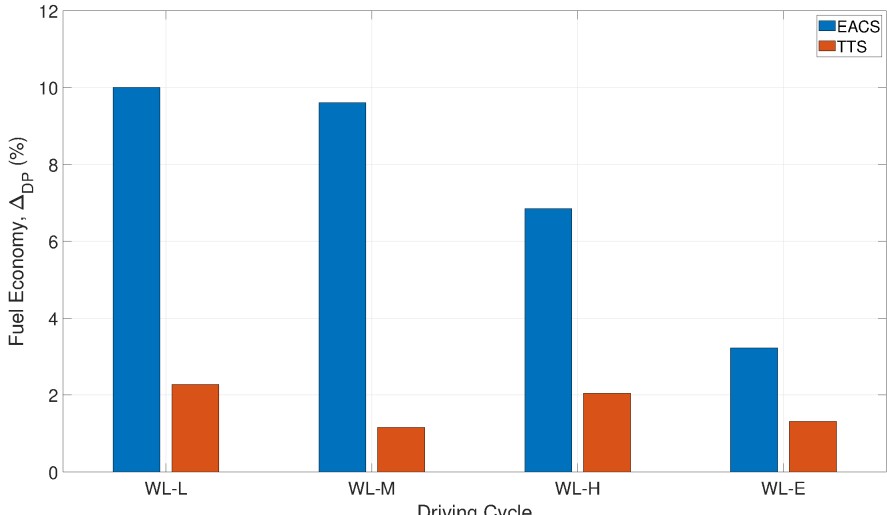

**Figure 8.** Comparison of fuel economy for the EACS and TTS relative to the performance of the DP, when driving the WL-L, WL-M, WL-H and WL-E [44].

Another advantage of TTS is that this strategy can be easily implemented for any kind of vehicle for which other control systems can not been easily implemented [44]. Lastly, a Simple TTS (STTS), described with just two parameters, can perform similar performance to TTS while it is more convenient to implemented in practice [44].

### 3.3.5. Deterministic Dynamic Programming

Dynamic Programming (DP) is a technique for solving a complicated problem by breaking it down into a collection of simpler sub-problems by means of the Bellman's "principle of optimality" [47]. DP method, according to the function to minimize (e.g., the fuel consumption of the vehicle), results in the identification of the optimal control strategy. In addition, DP allows to evaluate different layout designs of the vehicle in order to find the best results in terms of fuel consumption [46]. On the other hand, the algorithm uses a backward strategy, starting from the final value of the driving cycle, which must be known a-priori. In addition, this method is time consuming, thus making DP not suitable for the real-time control, but rather for off line optimization, as a benchmark for real-time strategies, unless it is coupled with predictive models [48].

In Reference [49] the dynamic programming is proposed as a method to optimize the cost function combining the fuel consumption and the selected emission species for the hybrid vehicle. In Reference [35] dynamic programming is applied in order to optimize the fuel economy and tracking safety for the adaptive cruise control. Besides, in Reference [50] dynamic programming and the Pontryagin's Minimum Principle (PMP) are compared for the energy management of HEVs. In Reference [51] a cost function, such as minimizing a combination of fuel consumption and selected emission species over a driving cycle, with the control constraints for the vehicle mode is defined and dynamic programming has been adopted to calculate the vehicle dynamics.

Considering the fact that DP strategy is time consuming, in Reference [31] a mathematical technique to speed up the calculation time has been proposed. The base principle of this method is in creating a matrix which includes outputs of the vehicle model (i.e., ICE chemical power and battery electric power). Since this matrix is evaluated only once before evaluating the optimal control strategy, there is no need to calculate it for each step, thus values can be easily obtained from the matrix for a given combination of the control variables. The most strong point of this method lies in the interaction among the model and permutations of control and status variables saved in the DP grid. Through this method the computational demand in MATLAB software is reduced since configuration matrix saves

all the information considering the vehicle in all probable scenarios, and it can be easily interfaced with the DP grid by means of simple matrix operations [31]. This method proposed to lessen the computational time for all the optimizers resulted 15 times faster than DP-based ones. This approach has been applied to two types of TTR-already introduced in Section 3.2—with different degrees of hybridization, TTR30 (30% hybridization) and TTR50 (50% hybridization). Results in terms of fuel economy reduction in respect to conventional vehicle are shown in Figure 9.

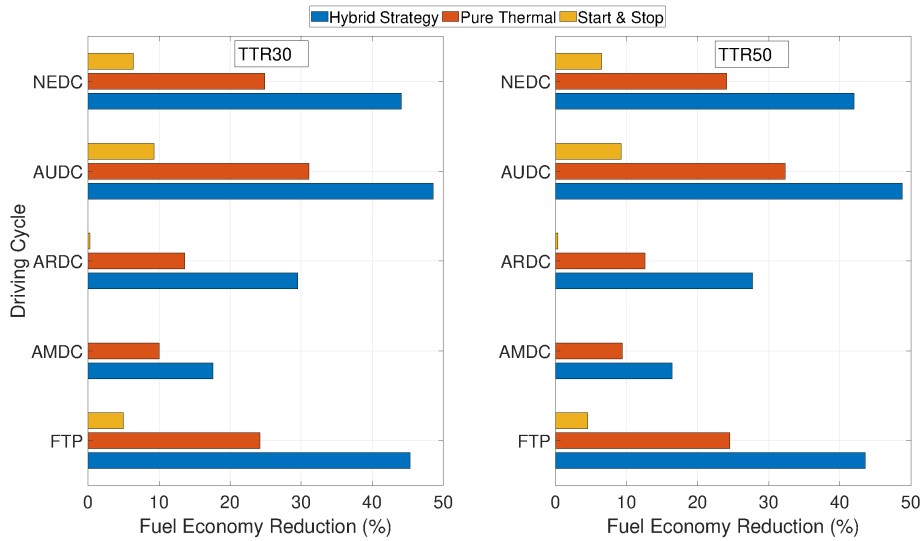

**Figure 9.** Fuel economy reduction with respect to the conventional vehicle, obtained with TTR30 and TTR50 vehicles for different driving cycles [31].

### 3.3.6. Cyber-Physical Predictive Energy Management

In Reference [52], a Cyber-Physical System (CPS) with a previous knowledge of road elevation to achieve the optimum fuel consumption was proposed. In the first part of the work, a TTR powertrain applied to a truck, whose architecture is shown in Figure 10, is proposed. In the second part, the proposed Average Power-based Model Predictive Controller (AP-MPC) is used for the energy management problem for this TTR hybrid vehicle with only road elevation known a-priori.

The characteristics of the main components for TTR hybrid vehicle are indicated in Table 10.

The prescient MPC requires all the future driving data such as velocity and road elevation, thus making it not suitable for real-time applications. Instead, the AP-MPC strategy only needs the road elevation-that can be available from vehicle connectivity-making it suitable for real-time applications [53].

AP-MPC is also compared with rule-based EMS, or rather those strategies for hybrid vehicles control that require limited data as vehicle torque, the battery SOC, brake demands, operating conditions of the electric motor and combustion engine efficiency and local torque/speed constraints. Two different paths (A, B) are investigated and are presented in Figure 11. Road elevation variations of path B are much larger than those of path A. This scenario is investigate to analyze the performance of AP-MPC in the mountain areas.

Performance achieved with the three different approaches are presented in Table 11.

The Rule-based EMS on the path A results in 9.5% saving on fuel consumption compared to the conventional vehicle. The prescient MPC improves the fuel economy by 18.5% with all the driving data known a-priori. However, the AP-MPC, which can be applied in real-time, results in a 16.9% improvement in fuel economy, which it is not far from the prescient MPC. Likewise, even on the path B, the AP-MPC strategy shows performance similar with the prescient MPC despite using less data than it.

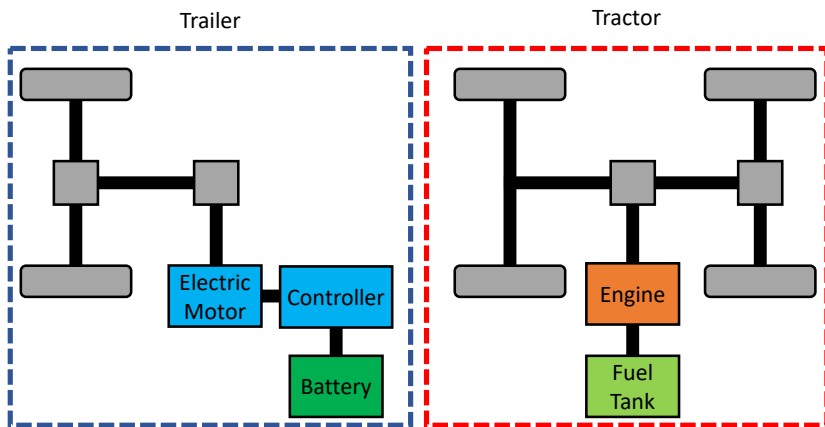

**Figure 10.** TTR truck powertrain schematic [52].

**Table 10.** Specification of TTR hybrid vehicle [52].

| Component | Main Parameters | Value |
|---|---|---|
| ICE | Displacement<br>Max Torque<br>Max Power | 14.9 l<br>1950 Nm @ 1200 rpm<br>368 kW |
| Battery | Capacity | 10 kWh |
| Motor | Max Torque<br>Rated Power<br>Rated Speed<br>Max Speed | 160 Nm<br>45 kW<br>3000 rpm<br>7000 rpm |
| Vehicle | Weight<br>Front Area | 36,700 kg<br>10 m$^2$ |
| Transmission | Number of gear ratio | 18 |

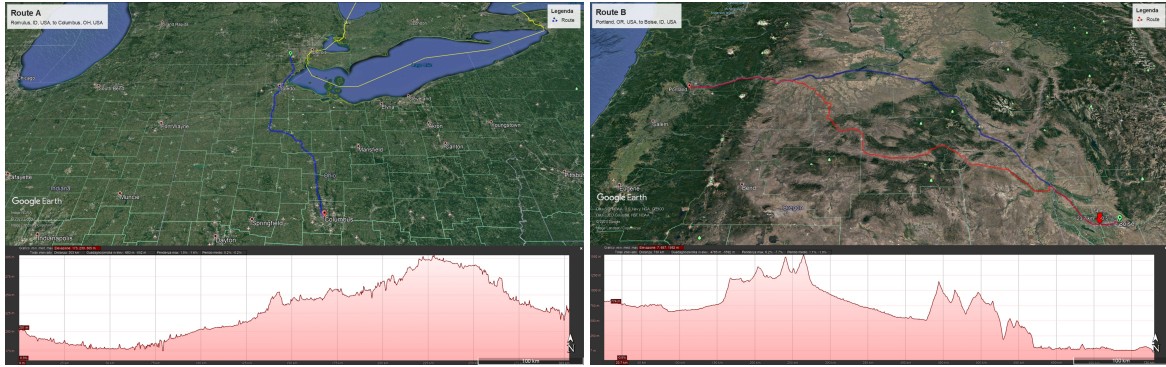

**Figure 11.** (**Left**) Path A: Romulus, MI, to Columbus, OH, USA (**Right**) Path B: Boise, ID, to Portland, OR, USA [52].

**Table 11.** Performance comparison among control strategies [52].

| TTR Vehicle | EMS | Driving Cycle | | | | | |
| | | Path A (648 km) | | | Path B (1394 km) | | |
| | | FC (l) | FS (%) | AFS (US$) | FC (l) | FS (%) | AFS (US$) |
|---|---|---|---|---|---|---|---|
| Hybrid | AP-MPC | 234.56 | 16.9% | 12,360 | 536.85 | 14.9% | 11,347 |
| | Prescient MPC | 230.10 | 18.5% | 13,516 | 530.27 | 15.9% | 12,140 |
| | Rule-based | 255.59 | 9.5% | 6910 | 588.22 | 6.8% | 5150 |
| Traditional | - | 282.25 | - | - | 630.91 | - | - |

In the margins of the analysis of vehicle performance when a control based on a cyber-physical system is adopted, it is necessary to mention that these systems lend themselves to possible attacks from the outside [54,55]. Automotive cybersecurity is a topic widely covered in the literature. In fact, the communication protocol for vehicles (CAN-Controlled Area Network) was not designed, back in the 1980s, making safety considerations. Consequently, the use of control units and devices connected externally via Bluetooth or other networks makes the vehicle's communication architecture vulnerable to external attacks, including DoS (Denial of Service), data extraction, code modification and interruption of functionality [56,57].

### 3.3.7. Sequential Linearization Control (SLC) methodology

In Reference [58], a Sequential Linearization Control (SLC) strategy is proposed. This strategy is based on an algebraic mapping of the accelerator pedal signal, the battery SOC, and the vehicle normalized speed into a single torque command for the electric motor. The ICE control utilizes an altered accelerator pedal to throttle plate angle using a tunable gain parameter that, in turn, determines the sustained battery SOC.

The SCL algorith was implemented for two PHEV operational strategies and for a HEV operational strategy:

- EVCD/HVCS: the vehicle starts in Electric Vehicle Charge Discharge (EVCD) mode with ICE turned off, depleting the battery down to a defined $SOC = SOC_{low}$. When $SOC_{low}$ is reached, the ICE is turned on and the vehicle operates in Hybrid Vehicle Charge Sustain (EVCS) mode.
- HVCD/HVCS: the vehicle starts in Hybrid Vehicle Charge Depleting (HVCD) mode, so with the ICE turned on, until $SOC_{low}$ is reached. At that level of charge, the operation mode turns into HVCS.
- HEV-CS: this operational strategies applies to a non-plugin HEV. Thus, the vehicle operates in Charge Sustain (CS).

The proposed methodology was validated using data acquired from an actual hybridized Ford Explorer in a TTR powertrain configuration, with an architecture depicted in Figure 1a. This vehicle was hybridized by a team of Georgia Institute of Technology students as an entry in three DOE/Ford FutureTruck university competition held between 2002 and 2004. The 3.0 liters V6 ICE drives the rear wheels of the vehicle through a five-speed automatic transmission, while front wheels are driven by a 150 kW AC-induction electric motor through a single fixed speed reducer. When the ICE is declutched from the rear differential, the vehicle drive in pure electric mode [59].

The simulated performance using EVCD/HVCS and HVCD/HVCS strategies with a 40 kWh battery and their comparison with actual on-road Explorer under HEV-CS control are reported in Table 12.

**Table 12.** Acceleration performance under EVCD/HVCS and HVCD/HVCS control strategies, and actual Explorer recorded data under HEV-CS strategy [58].

| Strategy | 0–60 mph Time (s) | Simulated 1/8 Mile Time (s) | Simulated 1/8 Mile Speed (mph) | Actual 1/8 Mile Time (s) | Actual 1/8 Mile Speed (mph) |
|----------|-------------------|-----------------------------|-------------------------------|--------------------------|------------------------------|
| EVCD/HVCS | 15.5 | 14.7 | 57.7 | - | - |
| HVCD/HVCS | 6.23 | 9.82 | 78.2 | - | - |
| HEV-CS | - | - | - | 10.6 | 66.0 |

Fuel consumption and SOC values at the end of driving schedules for the EVCD/HVCS and HVCD/HVCS strategy for different battery sizes are compared in Table 13.

**Table 13.** Comparison of fuel consumption and final SOC between strategies EVCD/HVCS and HVCD/HVCS for three battery sizes at the end of 16 cycles of each of the three driving schedules [58].

| Driving Schedule | Battery Size (kWh) | EVCD/HVCS (L/100km) | HVCD/HVCS (L/100km) | EVCD/HVCS (SOC) | HVCD/HVCS (SOC) |
|------------------|--------------------|--------------------|--------------------|-----------------|-----------------|
| US06 | 20 | 11.3 | 11.7 | 0.29 | 0.30 |
| | 40 | 8.9 | 10.2 | 0.29 | 0.37 |
| | 60 | 6.5 | 9.3 | 0.29 | 0.46 |
| UDDS | 20 | 7.4 | 9.4 | 0.34 | 0.39 |
| | 40 | 4.8 | 8.3 | 0.33 | 0.50 |
| | 60 | 1.3 | 7.8 | 0.31 | 0.59 |
| HWFET | 20 | 8.3 | 8.7 | 0.34 | 0.35 |
| | 40 | 6.3 | 7.5 | 0.33 | 0.40 |
| | 60 | 4.4 | 6.7 | 0.32 | 0.47 |

### 3.3.8. Two-Degree-of-Freedom LPV Control Via Torque Vectoring

A strategy proposed for solving the torque vectoring problem, i.e., the independent control of each wheel, of a TTR HEV is the Linear Parameter Varying (LPV) presented in [60]. In order to obtain high performance, especially under extreme driving situations, a two-Degrees-Of-Freedom (DOF) LPV self-scheduled controller is merged by using mixed sensitivity loop shaping. This strategy, which is based on a single Lyapunov function, can guarantee consistency and a level of control performance for all allowed trajectories.

This strategy has been implemented on a 14-DOF vehicle model for simulation. Among them, 6-DOF come from the center of gravity moving and rotating in all directions, 4-DOF are reserved for the suspension of the vehicle, and the other 4-DOF for the angular movement of the wheels.

The LPV controller is compared with a flatness-based controller where the generation of the force in longitudinal direction and the desired yaw momentum is demanded to a flat feedforward control in combination with a PID and a Linear Quadratic Gaussian (LQG) feedback control.

The proposed control was tested under severe driving situations and showed high and improved performance. It allowed to achieve good tracking and high robustness against disturbances and modeling errors.

### 3.4. Patents

A search on a popular patent database (patents.google.com) evidences a relatively limited but increasing number of patents focusing on TTR hybrid. Some of them are presented in the following.

### 3.4.1. Kit for Transforming a Conventional Motor Vehicle into a Solar Hybrid Vehicle, and Relevant Motorvehicle (Patent Number: WO/2011/125084)

This patent, registered by researchers of the University of Salerno (Italy), focuses on the development of equipments, along with associated techniques and methodologies, aimed at converting

conventional car into hybrid solar vehicles with TTR architecture [61]. Mild-hybridization is performed by installing in-wheel electric motors on the rear wheels and by the integration of photovoltaic panels on the roof. The original architecture was upgraded with an additional battery pack and a vehicle control unit to be faced with the engine management system by the OBD port, and not interfering with the original engine control unit (ECU). The modified vehicle architecture is depicted in Figure 12. The HySolarKit idea was first implemented converting a FIAT Grande Punto [19,25–27]. This project is now going to industrial phase, thanks to financing by Campania Region and by the EU Program LIFE and, with four Italian partners (www.life-save.eu).

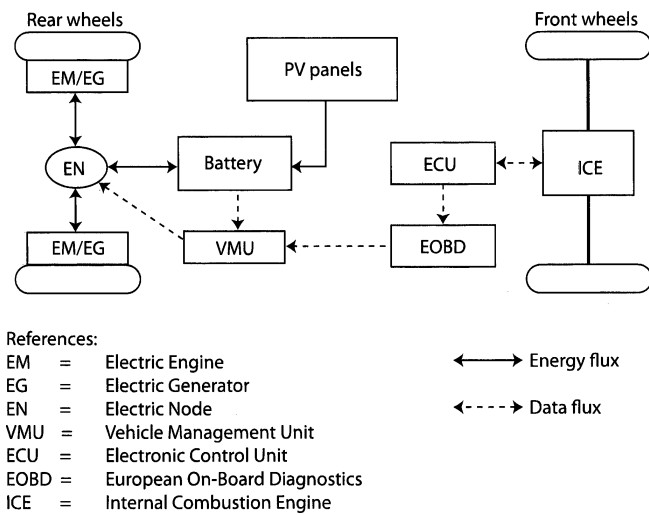

**Figure 12.** Block diagram of a vehicle modified by the system according to [61].

### 3.4.2. Electric Drive Controller Adaptation to Through-the-Road (TTR) Coupled Primary Engine and/or Operating Conditions (Patent Number: US2019/0202436 A1)

This patent, whose functional block diagram is illustrated in Figure 13, proposes a controller for trucks in TTR HEV configuration [62]. It uses control strategies such as an Equivalent Consumption Minimization Strategy (ECMS) or adaptive ECMS that are implemented at the supplemental torque delivering electrically powered drive axle (or axles) in a manner that follows operational parameters or computationally estimates states of the primary drivetrain and/or fuel engine, without participating in control of the internal combustion engine.

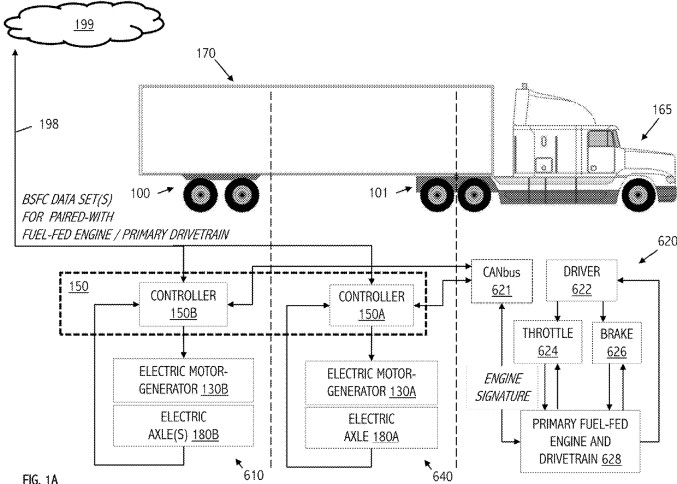

**Figure 13.** Functional block diagram illustrating hybridizing control [62].

### 3.4.3. Hybrid Vehicle Control Apparatus (Patent Number: US 6617704 B2)

This patent is focused on a hybrid vehicle control apparatus that controls an engine that drives at front wheels and a mechanically independent electric motor that moves at rear wheels [63]. A schematic view of the system is depicted in Figure 14.

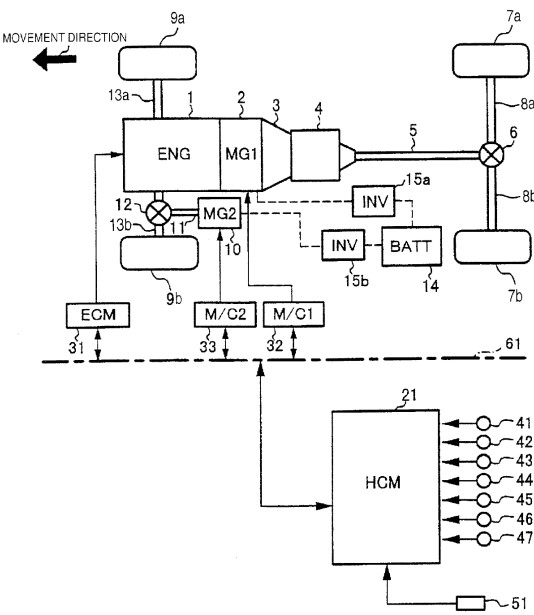

**Figure 14.** Schematic view of a drive transmission system of a vehicle equipped with a hybrid vehicle control apparatus according to [63].

The hybrid vehicle control apparatus stabilizes the vehicle driving performance when drive forces are obtained from both the engine and the electric motor. The control system stabilizes 4WD over a wide range using an engine and a comparatively small electric motor. The front (motor-driven) wheels drive torque are calculated by subtracting the rear (engine-driven) wheels drive torque from the total target drive torque.

The motor torque upper limit value is estimated from the motor rotational speed. If the front wheel target drive torque exceeds the motor torque upper limit value, the drive torque $\Delta T$, which corresponds to the amount by which the motor torque is insufficient, is added to the rear wheels (engine-driven) drive torque.

### 3.4.4. Multimodal Coupling for a Hybrid TTR Vehicle (Patent Number: DE112015003345T5)

A multimodal clutch is adapted to selectively connect and disconnect front and/or rear axles with respective engine and electric motor drivetrains connected to front and rear drive axles in a hybrid TTR vehicle [64].

The internal combustion engine is part of a front axle drivetrain connected to the front wheels, while the electric motor is part of a separate rear axle drivetrain connected to the rear wheels or vice-versa. By selective separation of an axle that is not actively driven, a reduction of parasitic losses in real time can be achieved, which can lead to an overall higher operational efficiency.

The multimodal coupling offers greater flexibility compared to the use of conventional friction clutches; each multimodal clutch can offer four different modes of operation to enable a wide variety of drive conditions. For example, freewheeling of the rear axle may occur in both directions of rotation whenever the electric motor is not in use. Possible embodiment of the system are shown in Figure 15.

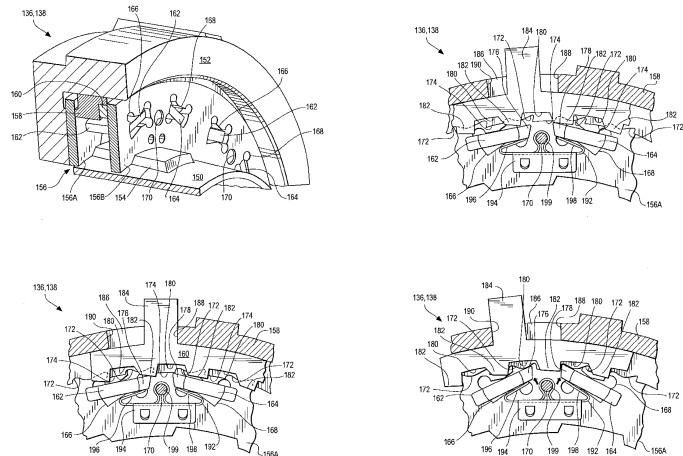

**Figure 15.** Views of possible embodiment of a multimodal clutch [64].

## 4. Conclusions

In parallel with the increasing diffusion of Hybrid Eletric Vehicles (HEV), also the Through-the-Road (TTR) solution is gaining popularity. Far from being a naive solution and in spite of some technical and architectural limitations, TTR HEVs are now being adopted for some models by major car manufacturers. An important advantage of this architecture is the prospect to retrofit conventional ICE vehicles and transform them into HEVs. This can represent a feasible short-term and low cost solution to convert part of the today car fleet, so contributing to reduce fuel consumption and emissions, avoiding a premature massive scrapping of conventional cars.

The literature study evidences an increasing effort both on patent side and on research activity on TTR HEVs, particularly on energy management, control strategies, vehicle and powertrain architecture and car conversion. This solution may have further development in the near future, particularly if a shortage of financial resources devoted to a fast transition toward fleet electrification would occur.

**Author Contributions:** Conceptualization, G.R. and F.A.T.; Funding acquisition, G.R.; Investigation, S.N. and M.M.; Project administration, G.R. and M.M.; Supervision, G.R. and F.A.T.; Visualization, F.A.T.; Writing–original draft, F.A.T., S.N. and M.M.; Writing–review & editing, F.A.T. and G.R. All authors have read and agreed to the published version of the manuscript.

**Funding:** This study is supported by a grant from European Union (LIFE-SAVE Solar Aided Vehicle Electrification LIFE16 ENV/IT/000442).

**Conflicts of Interest:** The authors declare no conflict of interest.

## Abbreviations

The following abbreviations are used in this manuscript:

HEV     Hybrid Electric Vehicle
ICE     Internal Combustion Engine
IWM     In-Wheel Motor
OBD     On-Board Diagnostic
PHEV     Plug-in Hybrid Electic Vehicle
TRL     Technology Readiness Level
TTR     Through-the-Road

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
