# Peer review of "A Survey on Through-the-Road Hybrid Electric Vehicles"

_electronics, doi:10.3390/electronics9050879_

Round 1

Reviewer 1 Report

Comments to the authors

Manuscript number: electronics-792736

Title: A Survey on Through-the-Road Hybrid Electric Vehicles

1) “hybrid-parallel” or “parallel-hybrid” or “parallel hybrid”? Concepts and terms must be consistent throughout the text.

2) The acronym “HEV” without being defined. Make sure all acronyms are defined properly before use.

3) Provide a brief outline of the manuscript at the end of Section 1. Moreover, Section 1 is somewhat incomplete. Last paragraph tells the history of hybrid vehicles, without any relation to the following section.

4) In abstract, only two configurations are discussed, while in Section 2 three configurations are discussed.

5) “Another advantage of this architecture is the …”. Since it is a new section, the authors cannot use pronouns like “this”. In this sentence, “this” is vague.

6)  In sections 3.1.1 and 3.1.2, mention the countries as well.

7) Any comments on the value of SOCL and SOCU? In microgrid applications, respectively 20% and 80% are used to determine the operation mode [R1]-[R2]. What about HEV?

[R1]. “Power management of an isolated hybrid AC/DC micro-grid with fuzzy control of battery banks,” IET Renewable Power Generation, 2015.

[R2]. “An SOC-Based Battery Management System for Microgrids,” IEEE Transactions on Smart Grid, 2014.

8) Provide the membership functions of the fuzzy logic presented in Table 5. Discuss on the shape and critical values of the membership functions.

9) One technical issue in cyber-physical systems is anomaly cyber attacks. See [R3]-[R4] for more details. Has this issue studied in the literature?

[R3]. “Optimal Data Injection Attacks in Cyber-Physical Systems,” IEEE Transactions on Cybernetics, 2018.

[R4]. “Feasibility and Detection of Replay Attack in Networked Constrained Cyber-Physical Systems,” in Proc. 57th Annual Allerton Conf. Communication, Control, and Computing, 2019.

10) Place the abbreviation section before Section 1.

Author Response

Dear review, thanks for your comments. They helped improving the paper. Please find below our answers:

1) Consistent terms have been checked. You will now find “parallel hybrid” wording.

2) The acronym HEV is defined in Page 1 Line 31, as it is used for the first time in the document. We added the definition in the abstract as well, in order to introduce it in that part of the document too.

3) An outline of the manuscript has been added at the end of Section 1 as you required. Another short paragraph that ties the introduction to the rest of the paper has been added. The paragraph on the history of hybrid vehicles, though interesting and oddity, has no relation with the following sections, so we decided to remove it.

4) The abstract has been corrected.

5) We fixed this part with “Another advantage of the TTR HEV architecture”

6) Countries have been added

7) HEV batteries operates in the range from 20 to 90% of the SOC in order to preserve battery life and safety. This consideration has been added in the manuscript.

8) An analysis of the fuzzy membership functions has been added.

9) Vulnerability of connected vehicles to cyber attacks has been studied in the literature. A brief consideration on this has been added in the manuscript.

10) The template provided by the journal put the abbreviation section at the end of the paper. Editor have the faculty to move it for the final release of the paper. Thus, we believe we should not move it at this stage.

Reviewer 2 Report

The authors have reviewed several types of TTR hybrid electric vehicles. The paper is well written and contains good material. However, the reviewer suggests the author should add block diagrams in different sections for comparison of various architectures and schemes. It will increase the readability of the paper. Moreover, the language of the paper should be carefully proofread.

Author Response

Dear reviewer. Thanks for your comments. They helped improving the paper.

The majority of researchers used the same architectures showed in Figure 1. To improve the reading of the paper we put a reference to Figure 1a or Figure 1b depending on what that researcher analyzed. If the work was about a different layour, we added specific block diagrams as you can see in Figures 7 and 10.

Round 2

Reviewer 1 Report

The manuscript is acceptable in the current form.